# Soft Polymer Optical Fiber Sensors for Intelligent Recognition of Elastomer Deformations and Wearable Applications

**DOI:** 10.3390/s24072253

**Published:** 2024-04-01

**Authors:** Nicheng Wang, Yuan Yao, Pengao Wu, Lei Zhao, Jinhui Chen

**Affiliations:** 1Institute of Electromagnetics and Acoustics and Key Laboratory of Electromagnetic Wave Science and Detection Technology, Xiamen University, Xiamen 361005, China; wangnicheng@stu.xmu.edu.cn (N.W.); 34320221150230@stu.xmu.edu.cn (P.W.); 15090173667@163.com (L.Z.); 2School of Informatics, Xiamen University, Xiamen 361005, China; furtheryao@stu.xmu.edu.cn

**Keywords:** polymer optical fiber sensors, wearable sensors, machine learning, multi-dimensional sensing, integrated system

## Abstract

In recent years, soft robotic sensors have rapidly advanced to endow robots with the ability to interact with the external environment. Here, we propose a polymer optical fiber (POF) sensor with sensitive and stable detection performance for strain, bending, twisting, and pressing. Thus, we can map the real-time output light intensity of POF sensors to the spatial morphology of the elastomer. By leveraging the intrinsic correlations of neighboring sensors and machine learning algorithms, we realize the spatially resolved detection of the pressing and multi-dimensional deformation of elastomers. Specifically, the developed intelligent sensing system can effectively recognize the two-dimensional indentation position with a prediction accuracy as large as ~99.17%. The average prediction accuracy of combined strain and twist is ~98.4% using the random forest algorithm. In addition, we demonstrate an integrated intelligent glove for the recognition of hand gestures with a high recognition accuracy of 99.38%. Our work holds promise for applications in soft robots for interactive tasks in complex environments, providing robots with multidimensional proprioceptive perception. And it also can be applied in smart wearable sensing, human prosthetics, and human–machine interaction interfaces.

## 1. Introduction

Soft robotics are gaining growing interest for their great potential in biomedical devices and robotic performance in extreme environments [1]. For example, soft robots enable compliance to highly confined spaces or self-locomotion on complex terrain [1,2,3]. In recent years, soft robots have evolved from traditional one-dimensional open-loop retractable actuators to three-dimensional closed-loop powerful execution devices [1,3,4]. For a soft robot to interact with its external environment, the perception ability of its own deformations via the sensors is essential. Traditional robotics mainly utilize resistive and piezoelectric type sensors, known for their high sensitivity and stability, and mature fabrications [5,6,7,8,9]. However, most of these electrical sensors struggle to operate in harsh industrial conditions that involve humidity and chemical corrosion, and they also face challenges in prolonged operations in complex electromagnetic environments [3,7,10,11]. Soft and wearable sensing devices based on optical waveguides (optical fibers) have emerged recently, and compared with their electronic counterpart, optical sensors have anti-electromagnetic interference and electrical capabilities [10,11,12,13,14,15,16,17,18,19,20]. In addition, optical sensing systems have the characteristics of a large transmission bandwidth, multiplexing, and diversified modulation/demodulation [11,21].

With respect to materials, the optical fiber sensors are mainly made of fused silica and polymers. The silica-type sensors usually show a high optical sensitivity, small insertion loss, and footprint. For example, Li et al. [13] reported a packaged hybrid plasmonic microfiber knot resonator for human health monitoring. They showed that the strain gauge factor was as large as 13,700, and the pressure sensitivity was 0.83 kPa^−1^. Zhang et al. [12] demonstrated ultrasensitive skin-like optical sensors based on a micrometer-scale fiber waveguide. They achieved an unprecedented pressure sensitivity of 1870 kPa^−1^, a low detection limit of 7 mPa, and a fast response time of 10 μs. Despite these recent advances, the conventional silica fiber sensors show a small dynamic range of mechanical deformation limited by the strength of the silica material, although extra structure design can improve the performance [11]. For soft robotic sensing applications, the optical sensors based on polymer materials have natural advantages since their mechanical modulus is close to human tissues, and they have better biocompatibility and stretchability [10,11,22,23]. Zhao et al. [14], reported an optoelectronically innervated soft prosthetic hand by using stretchable optical waveguides. They intentionally utilized high-loss materials to improve sensitivity during elongation. Guo et al. [24], demonstrated the highly flexible and stretchable strain sensors for human motion detection. They leveraged the wavelength-dependent absorption properties of the dye-doped polydimethylsiloxane (PDMS) fibers. Most of the soft sensors are designed to measure a certain type of deformation, and multimodal recognition usually relies on the high densities of sensors or complex analytical models [25,26,27]. Recently, Meerbeek et al. [25], proposed an internally luminescent elastomeric foam with proprioception. This soft sensory foam was trained using machine learning techniques to detect its deformation. This data-driven and machine-learning method overcomes the deficiency of conventional sensor decoding systems relying on physical modellings and can thus provide a new paradigm in advanced soft robotic sensors [17,28,29].

Here, we report a soft polymer optical fiber (POF) sensor for intelligent mechanical deformation recognition. The proposed optical sensing system is achieved by assembling the light-emitting diode (LED), POF sensors, and image sensors, as shown in Figure 1. We systematically study the sensor’s response to strain, bending, pressing, and long-term durability. By leveraging the intrinsic correlations of neighboring sensors and machine learning algorithms, we demonstrate the spatially resolved detection of the pressing and multimodal deformation of elastomers using a few POF sensors. Specifically, the developed intelligent sensing system can effectively recognize the two-dimensional (2D) indentation position with a prediction accuracy as large as ~99.17%. And the average prediction accuracy of the combined strain and twist is ~98.4%. Furthermore, as a proof-of-concept demonstration, we fabricate an integrated intelligent glove embedded with POF sensors for the recognition of hand gestures with a high recognition accuracy of 99.38%. The proposed optoelectronic sensors hold potential applications in soft robotics, wearable sensing, and human–machine interaction interfaces in the future.

## 2. Materials and Methods

Figure 1a illustrates the schematic diagram of the POF sensor device used for detecting multi-dimensional deformations of elastomer block, including bending, stretching, and torsion. The POF sensors array is embedded in a silicone elastomer (Figure 1b), which can support mechanical strength for sensors. The single POF sensor is fabricated as follows. First, we cut off a commercial POF of diameter 500 μm (polymethyl methacrylate, SC500-22 SUNMO, Shenzhen, China) with a cleaver. Then, we mechanically polish the output POF endface with a lapping film (1 μm grit, LF1P, Thorlabs, NJ, USA) as shown in Figure 1c. Finally, we connect the POFs in a rubber tubing and use ultraviolet (UV) glue (D-5604, ZhuoLide, Foshan, China) to seal the connection. The UV glue is cured with a 365 nm UV light source for ~30 s. Note that, in the experiment, it is observed that, when the strain reaches 200%, the curved UV glue will break off, and thus the repeatability of the sensors is spoiled. The air gap between the bridged POFs endface is initially set as ~700 μm. The small air gap is designed to enhance the deformation sensitivity of POF sensors, based on the fact that deformation frustrates the light transmission ability in the connected POF section [20]. The air gap distance is compromised by the detection sensitivity and the insertion loss. An LED serves as the light source, which is simply attached to a cleaved POF endface, and UV glue is utilized to fix the connecting section, as shown in Figure 1d. The emission of LED covers broad spectra in the visible region, and the typical coupling power into a POF is measured ~ 28 μW. The multi-POF bundle is assembled into a 3D-printed plastic block, and their real-time optical intensity responses are collected by an imaging camera (Figure 1a). Here, we use a high-resolution camera (MIchrome 5 Pro, Tucsen, Fuzhou, China) with a tube lens system to collect the sensors’ signals by default. For the integrated wearable sensors, a low-resolution image sensor (LT-USB802, Blue Sky Technologies, Shenzhen, China) is used to collect the sensors’ light signals. Each sensor’s output light intensity is extracted from the captured pictures. Specifically, the light intensity in a POF core region is extracted and summarized over the core region, which is regarded as the POF output intensity. Note that most previously reported soft optical sensors implement separate photodetectors or spectrometers to detect or decode the optical signals, and thus it would be challenging to scale up the massive sensing arrays and increase the cost [11,12,13,15,16,17]. In contrast, the proposed image-based detection can readily scale up the sensor network without an additional cost [25].

When the POF sensors are stretched or curved, the light coupling between the connected fiber endface is changed as shown in Figure 1e–g, resulting in modulating the transmitting power. For example, with applied stress, as depicted in Figure 1f, the rubber tube undergoes a certain deformation, increasing the distance between the fiber endfaces, and consequently, increasing the transmission light loss. Additionally, multiple POF sensors can be cascaded for distributed detections (Figure 1h). The maximal cascading sensors are limited by the insertion loss of the POF sensor unit. Thus, the proposed soft POF sensors can be used to detect the stretching, bending, and torsional deformation of the elastomer, as discussed later.

## 3. Results and Discussion

### 3.1. Quantification of POF Sensors to Mechanical Deformations

To quantify the optical sensitivity, stability, and dynamic response of the POF sensors, various tests are conducted. For the tensile sensing measurement, the POF sensor was fixed on a one-dimensional translation stage, and the strain was applied incrementally with a step size of ~14.28%, ranging from a relaxed state to a maximal strain of ~157.08%. As shown in Figure 2a, the normalized transmitted light intensity significantly decreases with increasing strain, and the repeatability is reasonably good for practical applications. We define strain sensitivity as *S* = Δ*I/ε*, where ∆*I* is the change in normalized intensity and *ϵ* is the change in strain [17,22]. Thus, the extracted strain sensitivity is approximately −0.2. The POF sensor was subjected to a cyclic tensile strain of 100% for 677 cycles, demonstrating stable strain responses (Figure 2b). In the strain durability test, one pigtail of the single POF sensor is fixed on the stage, and the other pigtail is fixed on an electronically controlled translation stage. The translation stage is programmed to periodically move forward and backward. The camera is also programmed to record the POF output light intensity synchronously. The dynamic response was also recorded during strain cycles, through continuous photography with a frame rate of 20 fps as shown in Figure 2c. Note that a much higher frame rate can be obtained from a commercial image sensor, while the 20 fps framerate should be suitable for mechanical motion detection.

As for bending sensing, the POF sensor was fixed on a rotation stage, and bending angles from 3° to 60° were applied with a step size of 3°. Figure 2d shows that the normalized output light intensity decreases with increasing bending angles, indicating good repeatability. It is found that the sensor is not quite sensitive to a small bending angle < 20°, which can be attributed to the large numerical aperture (0.5) of POF and the small gap distance [20]. In the linear bending response range (30–60°), the derived sensitivity is −2.1%/° and is comparable to the recently reported soft fiber sensors [20,22]. The bending sensitivity can be improved by simply tuning the air gap distance, although the effective angle range detection can be compromised [20]. The bending durability of the POF sensor was verified through 1000 bending cycles, as shown in Figure 2e. Dynamic responses during bending and recovery cycles were also measured, as shown in Figure 2f.

### 3.2. Demonstration of Spatially Resolved Pressing Recognition

The spatially resolved touch sensing has vibrant applications in human–machine interaction and soft robotic systems [1,2,30,31]. To this end, we fabricated a soft touchpad with an embedded sensor array, as shown in Figure 3a. Nine POF sensors were arranged to form a sensing area of 24 mm × 24 mm (black dash grid). The designed effective sensing area is based on the spatial indentation response of a single sensor, as shown in Figure 3b. Note that we fabricate a square elastomer embedded with a single sensor, as shown in Figure 3b inset. To obtain the spatial response map, we use a cylindrical rod with a diameter of ~ 2 mm as a probe to load on the elastomer and collect the output light signal simultaneously. The step distance of each measurement is set as 2 mm. We also measured the indentation response at four typical positions (Figure 3c). Generally, the POF sensor can only respond to an indentation depth larger than ~1.2 mm, which agrees with the buried depth of the sensors. In addition, the anisotropic indentation response is clearly observed, that is, the pressing along the axial direction shows a higher sensitivity than that of the orthogonal direction. Intuitively, the sensitive region distributes around the two POF endfaces. For 2D pressing recognition, a 13 × 13 grid is established with 2 mm spacing, and pressing is applied at each grid intersection, i.e., 169 detection points. Figure 3d shows that the first-row sensors (sensors ①–③) respond to the sequential indentation, i.e., X position No. = 1–13, Y position No. = 1. Since a single POF sensor has a certain detection area, there is a correlated response in the indentation motion. By harnessing the intrinsic correlations of neighboring sensors, it should be possible to realize distributed sensing with a small number of detection units. Here, as a proof-of-concept demonstration, we test the recognition of 169 (13 × 13) positions using a 3 × 3 sensors array, as indicated in Figure 3a. We collected 50 output fiber images per indentation position, and the Random Forest algorithm was used to process the datasets [32]. Figure 3e shows the 179 randomly selected sensing data from different press locations as inputs, with the counts indicating the number of sensing data at a certain position. Figure 3f shows the predicted results of the pressing positions and their counts. The predicted results agree well with the inputs, and thus the proposed intelligent sensing system can well recognize the 2D indentation position, and the prediction accuracy is ~99.17%.

### 3.3. Demonstration of Multimodal Deformation Recognition

The soft robotics is gaining research interest for their compliance to confined spaces or locomotion on uneven terrain [1,2,3]. For a soft robot to interact with its surroundings, the perception ability of its deformations via the sensors is required. Here, we leverage the correlated sensors array and machine learning to realize complex multi-dimension deformation, i.e., bend, stretch, and twist. To test the deformation recognition of bending/stretching and torsion, another strip-shaped elastomer embedded with sensors was fabricated, as shown in Figure 4a. The POF sensors array uses eight sensing components, arranged in an axial distance of 2 cm. The size of the entire elastomer was 105 mm × 20 mm× 8 mm. For multi-dimensional deformations, the elastomer was fixed between a one-dimensional translation stage and an electric rotating stage. The device was subjected to bending/stretching by manually tuning the displacement of the translation stage from −4 mm to 2.5 mm with a step of 0.5 mm. The typical bending state of the elastomer is illustrated in Figure 4b. Note that, here, the bending of elastomer is obtained by the negative displacement of the elastomer. The torsion angle is controlled by the rotating stage from −120° to 120° with a step of 30°. Figure 4c shows the images of twisted elastomers.

Similar to previous pressing recognition, here we again use the classical Random Forest algorithm to train the collected labeled dataset, and the schematic operation flow is shown in Figure 4d. In the training, 80% of the collected dataset is used to train the model, and 20% is used to validate. The average prediction accuracy of combined strain and twist is ~98.4%. We also compare the prediction accuracy among nine machine learning algorithms, including random forest, extra trees, KNeighbors, decision tree, MLP, SVC, AdaBoost, GaussianNB, and quadratic discriminant analysis [32,33,34,35,36], as shown in Figure 4e (see also Table 1). We find that random forest, extra trees, KNeighbors, and decision tree achieve a high accuracy rate exceeding 95%. Due to the advantages of easy-to-adjust hyperparameters, high accuracy, concise algorithm content, and fast operation speed, the random forest algorithm is used for verification in the subsequent verification. Figure 4e,f shows that the developed algorithm can well recognize the combined deformation states of elastomers.

### 3.4. Integrated Glove Sensors for Hand Gesture Recognition

The real-time detection of hand motions has many practical applications such as human–machine interaction, and prosthetic, or robotic hands design [3,6,28]. Here, we achieve an intelligent glove for real-time hand gesture recognition by combining soft POF sensors and random forest algorithm. We fabricate five soft stripe elastomers with embedded POF sensors, and then sew these elastomers onto the fingers of the glove, as shown in Figure 5a. The stripe soft sensors are obtained by putting the POF sensor into a 3D-printing plastic mold and then pouring the silicone gel for two hours precuring and half an hour heating (@ 80 °C). Like the previous measurements, LEDs are implemented as the light source, and at the receiving end, the output POF sensors are captured via a microscope imaging system. As a proof-of-concept demonstration, we focus on recognizing 16 typical hand gestures as shown in Figure 5a inset. Figure 5b shows the dynamic response of five POF sensors to various hand gestures. There are clear output light intensity changes for each hand gesture motion, which verifies the good sensitivity of the sensors for motion detection. We also test the cyclic response of sensors to a loop of gestures, i.e., sequential bending of thumb and index fingers, and their time-variable light intensity signals are shown in Figure 5c. The relatively good repeatability of the output light intensity change proves the robustness of the POF sensors.

In the training of gesture recognition datasets, we collect 16 gestures with labeled sensor responses. Each single gesture dataset contains 100 pictures; thus, a total of 1600 pictures was collected. We used 80% of the total dataset to train the model, and the remaining 20% to validate. When training the model, we investigate the relationship between the accuracy of the model and the size of the training dataset. It is observed that, when the training dataset reaches 5% of the total data size, the model’s accuracy has already reached 96% (Figure 5d), indicating the stability and robustness of the proposed sensing system. The results of validating the confusion matrix for 20% of the entire dataset are shown in Figure 5e. The results show that the accuracy of the gesture recognition glove reaches 99.38%.

For practical applications, it is important to integrate all the optical elements in a small footprint. In the aforementioned signal detections, we implement a bulky microscope imaging system to capture the output images of POF sensors. Here, we show that the signal detection can be realized by simply attaching the POF endface to an image sensor (LT-USB802), as shown in Figure 6a-b. The POFs are assembled into a 3D printing block, and the block is then directly attached to the image sensor, achieving an integrated, lightweight, and portable system. There are several advantages in such a detection configuration; first, the cost and footprint of the detection system are simultaneously reduced. Moreover, the robustness of the system is also improved without separate optical elements. The flow chart of the signal processing algorithm is shown in Figure 6b, including data collection, classifier training, and model inference. The guidelines for using such intelligent gloves are as follows. The user puts on the glove and actions according to the selected gestures, and during this time, the system can automatically collect the hand gesture datasets and train the model. Then, the system can be used to recognize the newly input hand gestures. The real-time recognition of hand gestures is shown in Figure 6d–g. The developed algorithms can recognize hand gestures precisely, which can be readily applied to human–machine interactions. We envision that, with more datasets of hand gestures, the intelligent glove should be able to recognize and model the hand motions.

## 4. Conclusions

In summary, we propose a soft POF sensor for mechanical deformation recognition. We systematically study the sensors’ response to strain, bending, pressing, and long-term durability. By leveraging the intrinsic correlations of neighboring sensors and machine learning algorithms, we demonstrate the spatially resolved detection of pressing and multi-dimensional deformation of elastomers. The developed intelligent sensing system can well recognize the 2D indentation position with a prediction accuracy as large as ~99.17%. And, the average prediction accuracy of combined strain and twist is ~98.4%. Furthermore, we demonstrate an integrated intelligent glove for the recognition of hand gestures with a high recognition accuracy of 99.38%. Our work may provide a cost-effective solution to practical applications in human–machine interactions and soft robotic sensors.

## Figures and Tables

**Figure 1 sensors-24-02253-f001:**
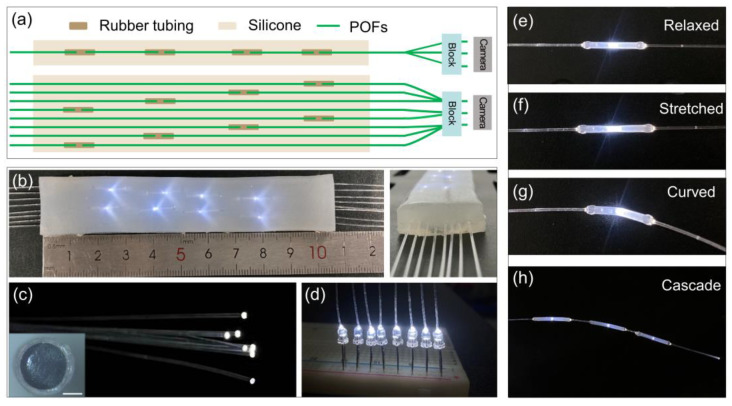
Polymer optical fiber (POF) sensors. (**a**) Schematic diagram of POF sensors array packaged in a silicone rubber. The output light signal is captured by a commercial image sensor. (**b**) The optical image of packaged POF sensors array in a silicone rubber with input light source. The scattering light spots indicate the embedded POF sensors. The right panel is the side view of the packaged POF device. (**c**) Image of POFs guiding light source. Inset is the microscope image of a polished POF endface. The scale bar is 200 μm. (**d**) Image of light-emitting diode (LED) sources connected to POFs. (**e**–**g**) Images of a POF sensor in the relaxed, stretched, and curved conditions, respectively. (**h**) Image of cascading POF sensors composing three units.

**Figure 2 sensors-24-02253-f002:**
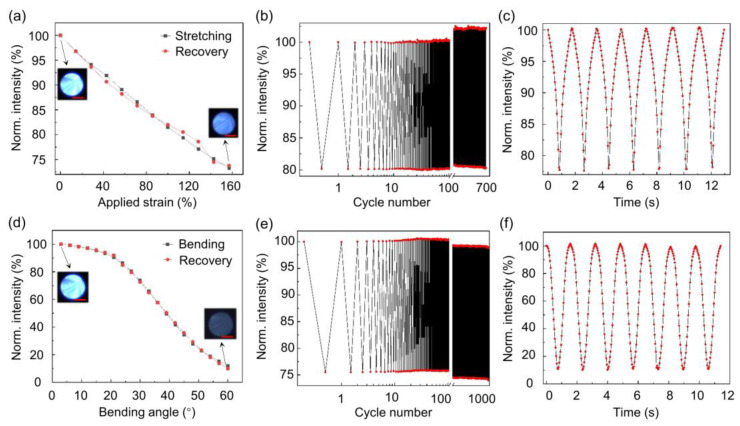
Characterizations of the mechanical sensing performance of POF sensors. (**a**) Normalized transmitted light intensity of a POF sensor related to the applied strain. The insets show the selected images of the output POF endface. The scale bar is 250 μm. (**b**) Straining durability test of the POF for a strain of 100%. (**c**) Dynamic responses of a POF sensor during stretching and recovery cycles. (**d**) Normalized transmitted light intensity of a POF sensor related to the bending angle. The insets show the selected images of the output POF endface. The scale bar is 250 μm. (**e**) Bending durability of the POF sensor at 1000 cycles. (**f**) Dynamic response of the POF sensor during bending and recovery cycles. The output light intensity is normalized by the maximal value of light intensity.

**Figure 3 sensors-24-02253-f003:**
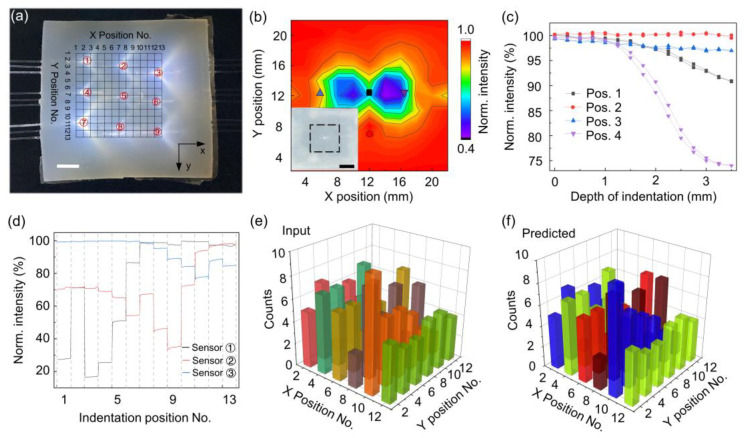
Demonstration of the spatially resolved detection of pressing. (**a**) Image of soft packaged sensors array composing nine POF sensors. The POF sensors are arranged into a 3 × 3 array (labeled with ①–⑨), and thus the sensing area (24 mm × 24 mm) is divided into smaller sensing regions of 13 × 13 (black dashed grid). The scale bar is 6 mm. (**b**) Spatial mapping of a single POF sensor response under a constant indentation depth of 3 mm. Inset is the optical image of a square elastomer embedded with a single POF sensor. The black dashed area is the testing region. The scale bar is 12 mm. (**c**) The indentation response for four typical positions. The measured positions are marked in (**b**). (**d**) The response of three embedded POF sensors (first row sensors) for sequential pressing along the y = 1 row. (**e**) The partial dataset of the spatial pressing response of POF sensors. (**f**) The Random Forest algorithm predicts the pressing position based on the sensing information of the sensor array.

**Figure 4 sensors-24-02253-f004:**
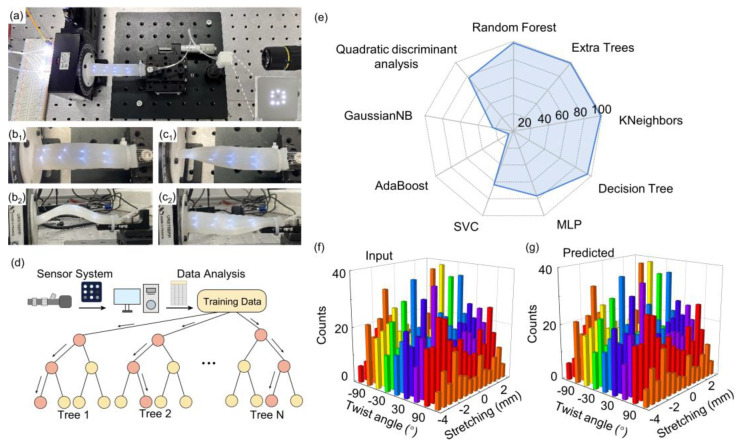
Multi-dimensional deformation detections. (**a**) Image of the experimental setup of POF sensors for multi-dimensional deformation detection. The inset is the output of POF sensors. (**b** (**b_1_**,**b_2_**)) Images of elastomer block with embedded POF sensors array in bent state (−4 mm displacement): (**b_1_**) top view; (**b_2_**) side view; (**c** (**c_1_**,**c_2_**)) images of elastomer block in torsion state (120° twist): (**c_1_**) top view; (**c_2_**) side view. (**d**) Schematic diagram of random forest algorithm. (**e**) Accuracy of different algorithm models for predicting the mechanical states (bend/stretch and twist) of the elastomer. (**f**) Input dataset of multi-dimensional deformation of the elastomer. (**g**) The random forest algorithm predicts the deformation states of the elastomer.

**Figure 5 sensors-24-02253-f005:**
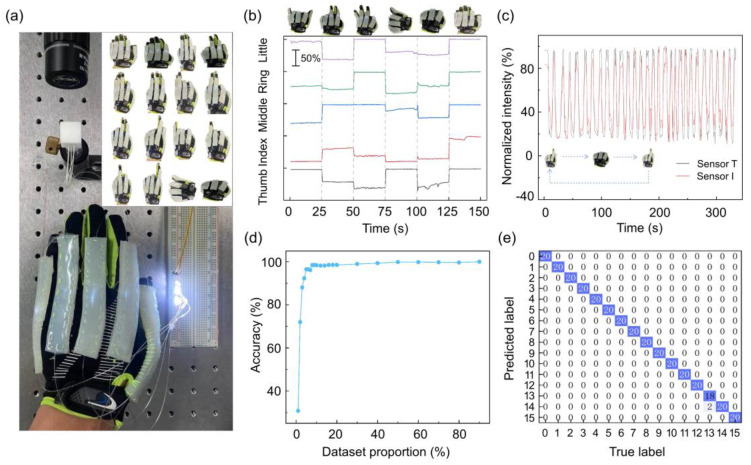
Demonstration of hand gesture recognition with POF sensors array. (**a**) Image of glove integrated with packaged POF sensors array and detection system. The output POF arrays are collected by a microscope image system. The inset shows the images of 16 different hand gestures for recognition. (**b**) The dynamic response of four-finger POF sensors under different gestures. The sensors are labeled with their deposition figures. (**c**) The cyclic stability of two POF sensors under repeated hand motions. Sensor T: thumb, sensor I: index. (**d**) The relationship between the accuracy of the training model and the size of the training dataset. (**e**) The confusion matrix of the experimental results. The blue box indicates the correct prediction counts.

**Figure 6 sensors-24-02253-f006:**
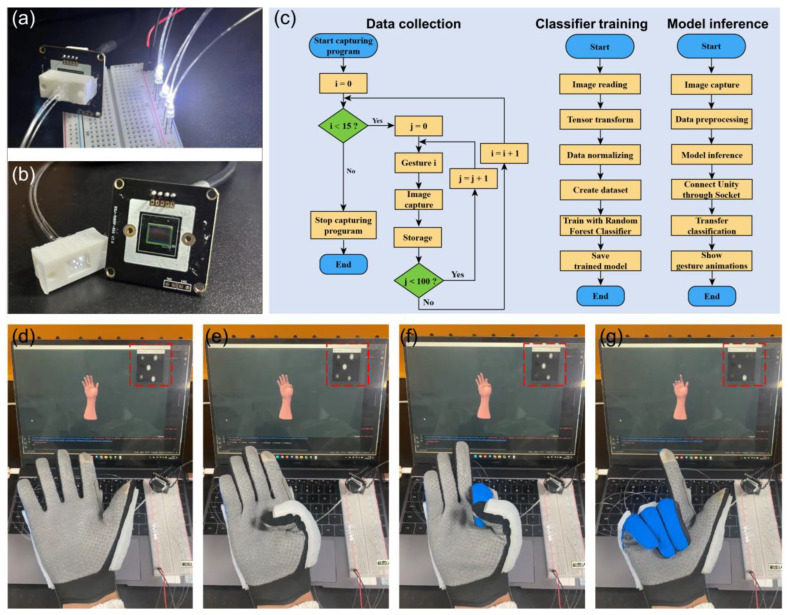
Intelligent hand gesture recognition with an integrated detection system. (**a**,**b**) Images of an integrated data acquisition system. The POF arrays are adapted into a printed block, which is then attached to a commercial image sensor. (**c**) Flow chart of real-time gesture recognition algorithms. The algorithms mainly contain three parts: data collection, classifier training, and model inference. (**d**–**g**) Demonstration of the real-time detection of hand gestures via the optical sensing system. The red-dashed box indicates the real-time output light field of POF sensors.

**Table 1 sensors-24-02253-t001:** Comparison of deformation recognition for different algorithms.

Algorithm	Prediction Accuracy for Strain (%)	Prediction Accuracy for Twist (%)	Prediction Accuracy for Combined Strain and Twist (%)
Random forest	99.13	99.71	98.37
Extra trees	99.86	99.95	98.85
KNeighbors	99.95	99.90	98.75
Decision tree	97.31	98.70	95.291
MLP	86.64	82.32	76.56
SVC	74.05	53.58	63.30
AdaBoost	33.68	26.81	5.96
GaussianNB	67.23	34.31	23.34
Quadratic Discriminant analysis	99.09	81.84	77.23

## Data Availability

Enquiries about data availability should be directed to the corresponding author.

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
