# Peer review of "Soft Polymer Optical Fiber Sensors for Intelligent Recognition of Elastomer Deformations and Wearable Applications"

_sensors, 2024, doi:10.3390/s24072253_

Round 1

Reviewer 1 Report

Comments and Suggestions for Authors

This paper demonstrates optical fiber sensors array for intelligent recognition of various types of deformations for use in the fields of wearable sensors and human-machine interaction. The design of the experiment is excellent and complete. This paper could be accepted after a minor revision.

The presentation of Figure 3 is difficult to understand. Since no clear sensors or data sources are given, it is difficult for readers to understand the author's testing methods and data processing methods.

1.      The designed effective sensing area is based on the spatial indentation response of a single sensor as shown in Figure 3b. Figure 3a includes 9 sensors? Which sensor does the mapping data in the Figure 3b come from?

2.      Figure 3c shows the data of indentation response at four typical positions. Where are the four typical positions?

3.      Which sensor do the three lines in the Figure 3d correspond to?

4.      Figure 3e and 3f need to give specific descriptions of processing methods. What does the Z axis as “counts” in the figure represent?

By the way, the author is suggested to propose a new title for this paper. The optical fiber sensor is not new. Or are arrayed design and identification methods new? The ultimate goal is practical application, not just for deformation testing of elastomers.

Reviewer 2 Report

Comments and Suggestions for Authors

Soft robotics are interesting in the field of biomedical devices and their robotic performance in various environments. POF sensors showed that their mechanical modulus is close to human tissues, and they have better biocompatibility and stretch ability. In this work, a soft POF sensor for intelligent mechanical deformation recognition, which gives a novel progress in this active field. Therefore, the work is recommended to be considered for publication for readers’ interesting. Before the publication, revision should be made as indicating  as following.

1In Materials and Methods, “…… polish the output POF endface with a lapping film as shown in Figure 1c” is stated, however, from the figure, there is no any information found concerning “polish” process and related results.

2 In “Thus, this setup facilitates the recognition of stretching, bending, and torsional deformation information of the elastomer, as discussed later.”, “this setup” should give more clear meaning.

3 In Materials and Methods, much more information is needed for measurements.

4 in Figure2, there is a need of more information for explanation of Figure2b, not a simple of “ (b) Straining durability test of the POF for a strain of 100%. “, otherwise, detailed description for the measurement is given in the part of Materials and Methods.

5 In Figure 6, data from the sensors’ measurements should be shown along with the hand gestures.

6 There is data for the conclusion of “we demonstrate an integrated intelligent glove for real-time recognition of hand gestures with a high recognition accuracy of 99.38%.”.

Reviewer 3 Report

Comments and Suggestions for Authors

The manuscript shows an AI-assisted sensor approach in recognition of system deformation. The proposed approach was integrated into wearable devices, demonstrating the capability to recognise hand gestures.

The work reported is scientifically sound and convincing. Nevertheless, I would like to suggest the following minor corrections to improve the quality of the manuscript even further.

- some typos have to be removed, and in general, I suggest an extensive language check.

- Figure 1 is in the wrong position. All figures should be first referred to in the text and then appear as close as possible to their first mention.

-Figures 5 and 6 appear too small for good readability in some parts. In particular, the representation of hand gestures in the inset in Figures 5a and 5b and the flow-chart of real-time gesture recognition.

-I suggest an accurate review of Section 2, which lacks most of the material detail. It is important to specify:

·       The UV-glue specification and the procedure of polymerization employed (exposure time and light source).

·       the LED characteristics (emission wavelength, power, and intensity)

·       the brand and the optical characteristics of the commercial imaging camera (reports all the different types of used cameras and justify their different use)

·       Specify the extraction procedure of the light intensity information by the captured image obtained from the camera.

·       Specify the procedure of normalization of light intensity

·       Specify the embedding procedure of the sensor system in the wearable device

-Could the UV-glue polymerization procedure influence the deformability of the POF system? Did you explore this phenomenon?

Round 2

Reviewer 2 Report

Comments and Suggestions for Authors

The new manuscript has revised according to the suggestions, I thisnk the work can be considered for publication.